# The role of financial stress, food insecurity, and COVID-19-related illness concerns shaping mental health in five South Asian countries during the pandemic (2020–2022): A secondary analysis of the online COVID-19 Trends and Impact Survey (CTIS) data

Youqi Yang[1]*, Lauren Zimmermann[2], Santanu Pramanik[3], Brian Wahl[4], Bhramar Mukherjee[5,6,7]

1 Department of Biostatistics, University of Michigan, Ann Arbor, Michigan, United States of America, 2 Department of Psychiatry, University of Michigan, Ann Arbor, Michigan, United States of America, 3 Economics and Statistics Unit, Centre for Effective Governance of Indian States (CEGIS), New Delhi, India, 4 Department of Epidemiology of Microbial Diseases, Yale University, New Haven, Connecticut, United States of America, 5 Department of Biostatistics, Yale University, New Haven, Connecticut, United States of America, 6 Department of Chronic Disease Epidemiology, Yale University, New Haven, Connecticut, United States of America, 7 Department of Statistics and Data Science, Yale University, New Haven, Connecticut, United States of America

* youqi@umich.edu

**Data availability statement:** The analysis code is publicly available at https://doi.org/10.5281/zenodo.15485043.

## Abstract

The COVID-19 pandemic has substantially impacted mental health worldwide, yet little attention has been given to its acute and long-term effects on mental health in low- and middle-income countries (LMICs). This study investigates how a triad of pandemic-related worries—financial stress, food insecurity, and COVID-19-related illness concerns—are associated with depression and anxiety across five South Asian LMICs: Bangladesh, India, Nepal, Pakistan, and Sri Lanka. Using data from the COVID-19 Trends and Impact Survey (CTIS), we analyzed responses from over 3.6 million participants collected between June 27, 2020 and June 25, 2022. We employed survey-weighted logistic regression models based on the complete cases (N = 1,062,786), adjusting for demographics and calendar time. Due to a substantial change in the survey design on May 20, 2021, our analysis was divided into two distinct periods: Period 1 (pre-change) and Period 2 (post-change). Our main findings reveal that all three types of pandemic-related worries were significantly associated with increased levels of depression and anxiety across the studied countries. In Period 1, a random-effects meta-analysis showed financial stress had the highest pooled adjusted odds ratio (OR)

**Funding:** The research was supported by the National Science Foundation (1712933 to BM). The funders had no role in study design, data collection and analysis, decision to publish, or preparation of the manuscript.

**Competing interests:** The authors have declared that no competing interests exist.

for depression at 2.41 (95% confidence interval, CI: [2.26, 2.58]), followed by COVID-19-related illness concerns at 1.58 (95% CI: [1.43, 1.75]), and food insecurity at 1.52 (95% CI: [1.40, 1.67]). In Period 2, the pooled adjusted OR for depression increased to 2.74 (95% CI: [2.38, 3.12]) for financial stress, while food insecurity showed a notable rise to 2.42 (95% CI: [2.23, 2.62]). Heterogeneity across countries was substantial ($I^2$ ranged from 60.33% to 86.68%), except for the association between food insecurity and depression in Period 2. Country-specific analyses further confirmed these results. Additionally, calendar time, vaccination status, gender, education, and rural-urban residential status modified these associations. These results underscore the need for targeted interventions to address socioeconomic stressors and improve mental health resilience in LMICs.

## Introduction

Depression and anxiety are among the most widespread mental health conditions globally [1]. Many researchers have investigated the subjective worries—such as financial stress, food insecurity, and health-related concerns—that are commonly associated with these conditions. A recent review of 40 observational studies found a consistent positive association between financial stress and depression [2]. In parallel, a global cross-sectional study using data from 149 countries reported that food insecurity was linked to poorer mental health [3]. During a public health crisis, such as infectious disease outbreaks, health-related illness concerns have been reported to be associated with elevated rates of psychological stress and disorders; this pattern was evident during the 2013–2016 Ebola outbreak in West Africa [4]. Examining potential social and economic effect modifiers of these associations can provide deeper insight into subgroup-specific vulnerabilities, as subjective experiences of worry often vary across individuals with different backgrounds, even when objective circumstances are similar. For instance, a study among U.S. college students found that the association between financial stress and symptoms of generalized anxiety was stronger among women than among men [5].

The dramatic spread of the COVID-19 pandemic profoundly affected not only physical but also mental well-being worldwide [6]. Researchers have consistently documented a rise in depression and anxiety among the general public during the pandemic [7–13]. A systematic review of 5,683 data sources estimated that the pandemic contributed to an additional 53.2 (95% confidence intervals, CI: [44.8, 62.9]) million cases of major depressive disorders and 76.2 (95% CI: [64.3, 90.6]) million cases of anxiety disorders worldwide [14]. Alongside the health crisis, the pandemic also severely disrupted economic systems and food supply chains. In response, a growing number of studies have explored how worries are associated with mental health outcomes in the context of COVID-19. Global evidence indicates that financial stress [15–20], food insecurity [21–27], and COVID-19-related health concerns [20,25,28,29] all played significant roles in the deterioration of mental health during this period. However, the data and evidence are not equitably distributed across all regions in the world.

This study focuses on five low- and middle-income countries (LMICs) from South Asia—Bangladesh, India, Nepal, Pakistan, and Sri Lanka—which together account for 23.5% of the global population. Systematic reviews have indicated increased burden of poor mental health outcomes among the general population in South Asia during the COVID-19 pandemic [30,31]. In response to the health crisis, all five countries implemented nationwide or regionally enforced strict public health interventions including lockdowns during 2020 and

2021 [32]. These disruptions to routine economic activities substantially constrained household resources, exacerbating poverty and contributing to increased food insecurity across the region [33–36]. Despite these documented impacts, there is limited quantitative research examining how pandemic-related worries are linked to mental health in the general adult population using comparable measurements and metrics across these countries over time. Most of the existing studies have focused on specific subpopulations within individual countries, such as rural women [37] and daily wage earners [38] in Bangladesh, small business entrepreneurs [39] and college students [40] in India, and urban informal sector laborers [41] in Pakistan. Most of these studies are cross-sectional and do not have repeated measures over the course of the pandemic. Furthermore, the literature on potential effect modifiers of the associations between worries and mental health remains very sparse. Where such analyses do exist, they have largely been conducted in high-income countries (HICs), including Canada [42], Portugal [43], and the United States [5,44], leaving a critical gap in understanding these dynamic interplays in South Asia and LMICs more generally.

Our study is a secondary analysis of the COVID-19 Trends and Impact Survey (CTIS), a large-scale global dataset covering a wide range of COVID-19-related topics from the onset through the later stages of the pandemic (2020–2022). CTIS has been widely recognized as a valuable resource for examining mental health outcomes [45–47] and their associations with pandemic-related worries [48,49], particularly in HICs. Researchers have leveraged this dataset both to study specific subpopulations—such as teachers [45]—and to analyze trends in the general adult population [47]. The availability of individual-level data, collected over time and across regions, makes CTIS especially valuable, and such granular and comparable data remain relatively rare in South Asia.

This study builds on our prior work in India [50] by expanding the scope to a regional analysis of five South Asian countries. We examine the associations between pandemic-related worries and depression/anxiety among the general population during the period from 2020 to 2022. By analyzing data across Bangladesh, India, Nepal, Pakistan, and Sri Lanka, we aim to uncover both commonalities and differences in these associations across countries that are geographically proximate yet diverse in terms of demographic profiles, socioeconomic conditions, healthcare systems, and pandemic responses [32,51–53]. In addition, we investigate how key socioeconomic factors—including vaccination status, self-reported gender, education, and rural-urban residential status—modify the relationship between worries and mental health. To our knowledge, this is the first cross-country analysis in LMIC settings to examine such effect modification, offering novel insights into the heterogeneity of mental health risks.

We hypothesize that significant associations exist between pandemic-related worries and both depression and anxiety across each of the studied countries. Furthermore, we anticipate variations in these associations over time and between countries, modified by socio-economic factors.

The structure of the rest of this paper is as follows: the "Materials and Methods" section describes the data source, measurement of variables, and statistical methods used in the analysis. The "Results" section presents both the exploratory and analytical findings. Finally, the "Discussion" section explores the implications of the findings, addresses the limitations of the study, and provides guidance for future pandemic preparedness informed by the analysis.

## Materials and methods

### Ethics statement

This study is a secondary analysis of the COVID-19 Trends and Impact Survey (CTIS) data in a de-identified format and is therefore exempt from Institutional Review Board review.

## Data source

We used individual-level data from the COVID-19 Trends and Impact Survey (CTIS), a large-scale, daily repeated cross-sectional survey. The survey was conducted by Meta/Facebook in collaboration with Carnegie Mellon University for the United States [54], and the University of Maryland for all other participating countries [55]. Participants were recruited via the Facebook platform, where a stratified random sample of monthly active users was invited to complete an online questionnaire. The survey collected detailed information on COVID-19-related topics, including mental health, pandemic-related worries, demographic characteristics, and more. To ensure the data's representativeness of the general adult population, Facebook applied a two-step weighting approach for each country [56]. First, non-response weights were derived using inverse propensity score weighting to mitigate non-response bias, improving the representativeness of survey participants within the active Facebook user base. Second, post-stratification weights were applied to align the sample's age and gender distributions with each country's general population, as reported by the United Nations (UN) Population Division's 2019 World Population Projections. This step accounted for non-coverage bias due to inactive Facebook users and individuals who do not use Facebook. The final weights applied in the analysis were the product of these two components. We accessed the de-identified individual response data and their associated weights for five South Asian countries through a Data User Agreement with the University of Maryland [57].

Our analysis spanned from June 27, 2020, the date when questions about pandemic-related worries were first introduced into the survey, to June 25, 2022, the final day of the survey. Our full sample included respondents who completed the entire questionnaire and reached the final screen (N = 3,644,631 across five South Asian countries). A major restructuring of the survey occurred on May 20, 2021 [58]. At this point, the survey adopted a modular format, introducing a set of core questions common to all respondents, along with two modules randomly shown to half of the participants each. Questions on mental health, financial stress, and food insecurity were included in only one of the modules, while the question on COVID-19-related health concerns was discontinued entirely due to less imminent severity concerns. Additionally, survey administrators noted a trend break in responses to core questions for the five South Asian countries following this restructuring [58]. To account for these changes, we divided our analysis into two distinct periods: Period 1 (June 27, 2020, to May 19, 2021; N = 2,083,979) and Period 2 (May 20, 2021, to June 25, 2022; N = 1,560,652). Each period was treated as a separate cross-sectional sample, independent of survey waves (S1 Figure).

## Measures

The data preparation procedures in this study adhere to a framework similar to that outlined in our previous paper [50]. We assessed two mental health measures, depression and anxiety, using binary variables to reflect the frequent presence of self-reported symptoms. These symptoms were measured with single-item adaptations of the Center for Epidemiological Studies Depression Scale (CES-D) and the Generalized Anxiety Disorder-7 (GAD-7), both widely validated tools for screening these conditions [59,60]. The survey items were introduced with the common prompt: "During the last 7 days, how often did you feel …?" Responses were originally recorded on a 5-point Likert scale and dichotomized for analysis. Participants who self-reported feeling the symptoms "all the time" or "most of the time" were coded as 1, while all other responses were coded as 0. This approach aligns with prior research using CTIS data [45–49]. It is important to note that these variables do not equate to clinical diagnoses, such as major depressive disorder or generalized anxiety disorder. Rather, the abbreviated single-item scales may be interpreted as the extent of self-reported feeling anxious or feeling depressed

within the past week. While the original responses for depression and anxiety showed good internal consistency (Cronbach's alpha = 0.79 in the full sample), we opted to report the two measures separately to maintain alignment with existing literature based on CTIS data [45–49].

We assessed three pandemic-related worries: financial stress, food insecurity, and COVID-19-related illness concerns. Financial stress was measured by asking: "How worried are you about your household's finances in the next month?" Food insecurity was assessed with the question: "How worried are you about having enough to eat in the next week?" COVID-19-related illness concerns were evaluated through the question: "How worried are you that you or someone in your immediate family might become seriously ill from coronavirus (COVID-19)?" Responses were originally recorded on a 4-point Likert scale: 1 = "very worried", 2 = "somewhat worried", 3 = "not too worried", 4 = "not worried at all". For analysis, responses were dichotomized into 1 for "very worried" and 0 for all other responses, consistent with earlier work conducted with CTIS data [48,49].

We also considered demographic variables, including gender, age, education, rural-urban residential status, occupation, and vaccination status. Details on the data processing for these covariates are provided in S1 Appendix. The wording of both the survey questions and the original participant responses can be found in S1 Table. Additionally, we incorporated calendar time (month and year of survey collection) as a variable in our analysis to account for temporal trends.

## Statistical methods

For the exploratory analysis, we used the full sample from both periods. Summary statistics for several demographic variables (gender, age, education, and rural-urban residential status), worries about the pandemic, and mental health measures were calculated for each country, both unweighted and weighted. To address non-response and non-coverage bias, we applied weights provided by Facebook, as described in the previous section. For comparison, we included demographic distributions of the general adult population for each country, drawn from the UN Population Division's 2022 World Population Projections [61] and the World Bank Open Data [62].

For the statistical analysis, we conducted separate survey-weighted logistic regression models for depression and anxiety as outcomes, stratified by period and country, using the complete cases. Detailed descriptions of the statistical models are provided in S2 Appendix. The primary models included pandemic-related worries (financial stress, food insecurity, and COVID-19-related illness concerns) as exposures, with demographic variables (gender, age, education, rural-urban residential status, and occupation), and calendar time (categorized by month and year) as covariates. To assess the overall effect of pandemic-related worries across countries, we performed a random-effects meta-analysis using restricted maximum likelihood (REML), assuming heterogeneity in effects across nations. This meta-analysis pooled estimates for each worry-outcome association within each period. For this part of the analysis, complete cases were defined as observations with no missing values for the included exposures, covariates, and outcomes (N = 827,472 in Period 1 and N = 235,314 in Period 2).

We further explored effect modification in the associations between pandemic-related worries and mental health outcomes. Potential effect modifiers included vaccination status, gender, education, rural-urban residential status, and calendar time. For vaccination status, we extended our primary models by introducing vaccination status and its interaction

term with each pandemic-related worry separately, while adjusting for all previously mentioned exposures and covariates. Since national vaccination programs in the five study countries expanded vaccine eligibility to all adults toward the end of Period 1 (e.g., May 1, 2021, in India [63]), vaccination status was examined as a potential effect modifier only in Period 2. In this part of the analysis, complete cases were further restricted to individuals with non-missing vaccination status (N = 234,149).

For other potential effect modifiers (gender, education, rural-urban residential status, and calendar time), we fitted separate models that included interaction terms between each modifier and one pandemic-related worry at a time, adjusting for all previously mentioned exposures and covariates in the primary models. Complete cases for this analysis included observations with no missing values in the relevant exposures, covariates, and outcomes (N = 827,472 in Period 1 and N = 235,314 in Period 2).

Robust sandwich estimators were used to estimate variance, and statistical significance was evaluated using the Wald test. All analyses were conducted in R version 4.4.2, using the survey package [64] for survey-weighted models and the metafor package [65] for meta-analysis. The analysis code is publicly available at https://doi.org/10.5281/zenodo.15485043.

## Results

### Descriptive statistics

Table 1 presents the unweighted and weighted distributions of demographic variables (gender, age, education, and rural-urban residential status) for the full sample (N = 3,644,631). In comparison to census data for each country, the unweighted sample overrepresents males (71% to 86% across countries and periods vs. 48% to 52%), underrepresents individuals aged over 65 years (1% to 7% vs. 7% to 16%), overrepresents individuals with a high school degree or higher (80% to 94% vs. 13% to 64%), and underrepresents individuals who resident in rural areas (17% to 30% vs. 60% to 81%).

After applying weights, the distributions for gender and age—both considered in the weight calculations—align more closely with the census data. However, education and residential status, which were not included in the weighting process, remain largely imbalanced. This finding indicates potential bias persisting towards those with higher education and in urban settings, and is consistent with previous research [50]. Taking Pakistan in Period 2 as an example: the weighted proportion of males decreases from 81% unweighted to 54%, which is closer to the census value of 51%. However, for education, the weighted proportion of individuals with a high school degree or higher remained largely unchanged (93% unweighted vs. 95% weighted), substantially exceeding the census estimate of 27%. Similarly, the proportion of rural residents decreased from 19% unweighted to 15% weighted, further deviating from the census value of 62%.

For gender and age, the alignment between the weighted proportions and the census proportions varied across periods and nations. For example, similar to Pakistan during Period 2, the weighted proportion of males aligned relatively closely with the census data in both Period 1 (50% vs. 48%) and Period 2 (49% vs. 48%) both in Nepal and in Period 2 (49% vs 48%) in Sri Lanka. However, in other periods and countries, the weighted proportions of males did not align as well with the census, with the largest discrepancy observed in Period 1 in Bangladesh (69% vs 49%).

Table 2 shows the unweighted and weighted distributions of pandemic-related worries and mental health measures for the full sample (N = 3,644,631). Regarding pandemic-related worries, the weighted proportion of individuals expressing stress about financial stability ranged

**Table 1. Summary of unweighted (light green) and weighted (light blue) statistics for demographics (gender, age, education, and rural-urban residential status) in the full sample by country and period (N = 3,644,631), compared to census data (light orange).**

| | Bangladesh | | | | | India | | | | | Nepal | | | | | Pakistan | | | | | Sri Lanka | | | | |
|---|---|---|---|---|---|---|---|---|---|---|---|---|---|---|---|---|---|---|---|---|---|---|---|---|---|
| | Period 1 | | Period 2 | | Cen. | Period 1 | | Period 2 | | Cen. | Period 1 | | Period 2 | | Cen. | Period 1 | | Period 2 | | Cen. | Period 1 | | Period 2 | | Cen. |
| Sample size | 173,588 | | 225,911 | | | 1,532,776 | | 1,043,398 | | | 119,691 | | 80,461 | | | 207,442 | | 172,566 | | | 50,482 | | 38,316 | | |
| Variables | UW | W | UW | W | | UW | W | UW | W | | UW | W | UW | W | | UW | W | UW | W | | UW | W | UW | W | |
| **Demographics** | | | | | | | | | | | | | | | | | | | | | | | | | |
| **Gender** | | | | | | | | | | | | | | | | | | | | | | | | | |
| Female | 14% | 31% | 15% | 40% | 51% | 17% | 34% | 17% | 43% | 48% | 24% | 50% | 24% | 51% | 52% | 19% | 35% | 19% | 46% | 49% | 28% | 46% | 29% | 51% | 52% |
| Male | 86% | 69% | 85% | 60% | 49% | 83% | 66% | 83% | 57% | 52% | 76% | 50% | 76% | 49% | 48% | 81% | 65% | 81% | 54% | 51% | 72% | 54% | 71% | 49% | 48% |
| (Missing) | 66,318 | | 76,670 | | | 597,175 | | 334,133 | | | 40,339 | | 21,853 | | | 72,370 | | 51,570 | | | 15,545 | | 8,789 | | |
| **Age** | | | | | | | | | | | | | | | | | | | | | | | | | |
| 18-24 | 31% | 26% | 33% | 22% | 20% | 22% | 21% | 20% | 17% | 18% | 34% | 31% | 32% | 28% | 22% | 29% | 27% | 27% | 23% | 24% | 15% | 15% | 13% | 13% | 14% |
| 25-64 | 68% | 72% | 66% | 74% | 70% | 76% | 74% | 76% | 75% | 72% | 65% | 66% | 66% | 67% | 68% | 69% | 70% | 70% | 72% | 69% | 81% | 76% | 80% | 74% | 70% |
| 65+ | 1% | 2% | 1% | 4% | 10% | 2% | 5% | 5% | 8% | 10% | 1% | 3% | 2% | 5% | 10% | 2% | 3% | 3% | 6% | 7% | 4% | 9% | 7% | 13% | 16% |
| (Missing) | 64,466 | | 75,993 | | | 578,727 | | 328,421 | | | 38,682 | | 21,551 | | | 69,388 | | 50,698 | | | 14,751 | | 8,674 | | |
| **Education** | | | | | | | | | | | | | | | | | | | | | | | | | |
| HS or more | 80% | 79% | 91% | 92% | 29% | 83% | 83% | 92% | 93% | 30% | 87% | 86% | 91% | 90% | 13% | 94% | 94% | 93% | 95% | 27% | 89% | 89% | 91% | 91% | 64% |
| Less than HS | 20% | 21% | 9% | 8% | 71% | 17% | 17% | 8% | 7% | 70% | 13% | 14% | 9% | 10% | 87% | 6% | 6% | 7% | 5% | 73% | 11% | 11% | 9% | 9% | 36% |
| (Missing) | 73,036 | | 77,753 | | | 660,043 | | 337,634 | | | 43,134 | | 22,162 | | | 74,435 | | 51,983 | | | 16,935 | | 8,973 | | |
| **Residential status** | | | | | | | | | | | | | | | | | | | | | | | | | |
| Rural | 21% | 23% | 25% | 22% | 60% | 26% | 25% | 29% | 25% | 64% | 20% | 18% | 21% | 17% | 79% | 17% | 16% | 19% | 95% | 62% | 30% | 31% | 29% | 28% | 81% |
| Urban | 79% | 77% | 75% | 78% | 40% | 74% | 75% | 71% | 75% | 36% | 80% | 82% | 79% | 83% | 21% | 83% | 84% | 81% | 5% | 38% | 70% | 69% | 71% | 72% | 19% |
| (Missing) | 67,683 | | 78,130 | | | 615,717 | | 340,369 | | | 41,789 | | 22,380 | | | 73,978 | | 52,357 | | | 16,113 | | 9,087 | | |

Abbreviations: Period 1, June 27, 2020, to May 19, 2021; Period 2, May 20, 2021, to June 25, 2022; UW, unweighted; W, weighted; Cen., census; HS, high school. Census data on gender, age, and residential status distributions were sourced from the UN Population Division's 2022 World Population Projections [61], while education distributions were obtained from World Bank Open Data [62]. Note: a) Survey participants self-reported gender information, whereas census data represents biological sex.

**Table 2. Summary of unweighted (light green) and weighted (light blue) statistics for worries about the pandemic and mental health measures in the full sample by country and period (N = 3,644,631).**

| | Bangladesh | | | | India | | | | Nepal | | | | Pakistan | | | | Sri Lanka | | | |
|---|---|---|---|---|---|---|---|---|---|---|---|---|---|---|---|---|---|---|---|---|
| | Period 1 | | Period 2 | | Period 1 | | Period 2 | | Period 1 | | Period 2 | | Period 1 | | Period 2 | | Period 1 | | Period 2 | |
| Sample size | 173,588 | | 225,911 | | 1,532,776 | | 1,043,398 | | 119,691 | | 80,461 | | 207,442 | | 172,566 | | 50,482 | | 38,316 | |
| Variables | UW | W | UW | W | UW | W | UW | W | UW | W | UW | W | UW | W | UW | W | UW | W | UW | W |
| **Worries about the pandemic** | | | | | | | | | | | | | | | | | | | | |
| **Financial stress** | | | | | | | | | | | | | | | | | | | | |
| Not worried | 74% | 76% | 76% | 78% | 77% | 79% | 80% | 83% | 78% | 79% | 81% | 84% | 79% | 79% | 80% | 82% | 77% | 80% | 80% | 82% |
| Worried | 26% | 24% | 24% | 22% | 23% | 21% | 20% | 17% | 22% | 21% | 19% | 16% | 21% | 21% | 20% | 18% | 23% | 20% | 20% | 18% |
| (Missing) | 58,504 | | 154,579 | | 504,130 | | 694,804 | | 33,735 | | 51,485 | | 62,474 | | 113,042 | | 12,511 | | 23,417 | |
| **Food insecurity** | | | | | | | | | | | | | | | | | | | | |
| Not worried | 89% | 89% | 88% | 89% | 92% | 92% | 92% | 92% | 93% | 93% | 93% | 94% | 92% | 92% | 91% | 92% | 89% | 90% | 91% | 92% |
| Worried | 11% | 11% | 12% | 11% | 8% | 8% | 8% | 8% | 7% | 7% | 7% | 6% | 8% | 8% | 9% | 8% | 11% | 10% | 9% | 8% |
| (Missing) | 58,686 | | 154,500 | | 505,849 | | 694,892 | | 33,920 | | 51,492 | | 62,766 | | 112,891 | | 12,567 | | 23,507 | |
| **COVID-19-related illness concern** | | | | | | | | | | | | | | | | | | | | |
| Not worried | 70% | 71% | | | 75% | 76% | | | 74% | 74% | | | 72% | 72% | | | 58% | 60% | | |
| Worried | 30% | 29% | | | 25% | 24% | | | 26% | 26% | | | 28% | 28% | | | 42% | 40% | | |
| (Missing) | 59,449 | | | | 510,338 | | | | 34,689 | | | | 63,951 | | | | 12,945 | | | |
| **Mental health** | | | | | | | | | | | | | | | | | | | | |
| **Depression** | | | | | | | | | | | | | | | | | | | | |
| Yes | 15% | 15% | 17% | 17% | 8% | 8% | 9% | 8% | 7% | 7% | 9% | 8% | 11% | 12% | 13% | 14% | 9% | 9% | 12% | 12% |
| No | 85% | 85% | 83% | 83% | 92% | 92% | 91% | 92% | 93% | 93% | 91% | 92% | 89% | 88% | 87% | 86% | 91% | 91% | 88% | 88% |
| (Missing) | 54,095 | | 152,575 | | 450,298 | | 681,320 | | 30,527 | | 50,594 | | 57,422 | | 111,562 | | 11,359 | | 23,189 | |
| **Anxiety** | | | | | | | | | | | | | | | | | | | | |
| Yes | 10% | 11% | 12% | 11% | 5% | 6% | 7% | 6% | 5% | 5% | 7% | 7% | 8% | 9% | 10% | 9% | 7% | 7% | 10% | 10% |
| No | 90% | 89% | 88% | 89% | 95% | 94% | 93% | 94% | 95% | 95% | 93% | 93% | 92% | 91% | 90% | 91% | 93% | 93% | 90% | 90% |
| (Missing) | 54,120 | | 152,161 | | 450,656 | | 680,232 | | 30,823 | | 50,392 | | 57,776 | | 111,045 | | 11,528 | | 23,159 | |

Abbreviations: Period 1, June 27, 2020, to May 19, 2021; Period 2, May 20, 2021, to June 25, 2022; UW, unweighted; W, weighted. COVID-19-related health concerns were excluded from the surveys during Period 2.

from 16% to 24% across the five South Asian countries. Weighted estimates for food insecurity ranged from 6% to 11%. Concerns about contracting COVID-19 were particularly prominent in Sri Lanka, where the weighted proportion was 40%, compared to a range of 24% to 29% in the other four countries. For mental health measures, the weighted proportions of individuals self-reporting depression varied between 7% and 17%, while anxiety ranged from 5% to 11%. Bangladesh reported the highest weighted proportions for both depression (15% in Period 1 and 17% in Period 2) and anxiety (11% in Period 1 and 11% in Period 2) during both periods, standing out among the five South Asian countries.

## Associations between pandemic-related worries and mental health

Fig 1 displays the unpooled and pooled odds ratios (ORs) and 95% CIs for the associations between pandemic-related worries and two mental health measures (depression and anxiety), using the complete cases (N = 1,062,786). After adjusting for demographic variables (gender, age, education, rural-urban residential status, and occupation) and calendar time (categorized by month and year), significant associations were found between all pandemic-related worries and both depression and anxiety in the five South Asian countries, except for COVID-19-related illness concerns, which were not assessed in the survey in Period 2.

**Depression.**  In Period 1, pooled analysis identified financial stress as the strongest pandemic-related factor associated with depression, with individuals experiencing high intensity of financial stress demonstrating 2.41 times the odds (95% CI: [2.26, 2.58]) of frequent depression compared to those with low, moderate, or no financial stress. Food insecurity and COVID-19-related illness concerns also significantly increased depression risk, showing pooled adjusted ORs of 1.52 (95% CI: [1.40, 1.67]) and 1.58 (95% CI: [1.43, 1.75]), respectively. Considerable variability across countries was observed, as indicated by $I^2$ values, reflecting the percentage of variability explained by heterogeneity among countries: 60.33% for financial stress, 66.49% for food insecurity, and notably higher at 86.68% for COVID-19-related illness concerns.

Country-specific findings in Period 1 supported these pooled results, with adjusted ORs for depression among individuals experiencing high financial stress versus those with lower or no financial worries ranging from 2.18 (95% CI: [1.98, 2.41]) to 2.53 (95% CI: [2.43, 2.64]). For food insecurity, country-level adjusted ORs ranged between 1.26 (95% CI: [1.04, 1.53]) and 1.80 (95% CI: [1.49, 2.17]). COVID-19-related illness concerns produced adjusted ORs ranging from 1.34 (95% CI: [1.17, 1.53]) to 1.79 (95% CI: [1.56, 2.17]).

In Period 2, pooled estimates revealed strengthened associations between pandemic-related worries and depression, although with increased variability across countries, likely due to smaller sample sizes. Specifically, financial stress showed a pooled adjusted OR of 2.74 (95% CI: [2.38, 3.12]), while food insecurity yielded an elevated pooled adjusted OR of 2.42 (95% CI: [2.23, 2.62]). Heterogeneity remained high for financial stress ($I^2$=71.43%), though notably absent for food insecurity ($I^2$=0%).

Country-specific analyses in Period 2 highlighted variability in the strength of these associations. For example, in certain countries such as Nepal, the adjusted OR for depression due to financial stress notably increased from 2.38 (95% CI: [2.04, 2.77]) in Period 1 to 3.68 (95% CI: [2.80, 4.84]) in Period 2.

**Anxiety.**  In Period 1, financial stress exhibited the strongest pooled association, with an adjusted OR of 1.87 (95% CI: [1.69, 2.08]). Food insecurity and COVID-19-related illness concerns also showed notable associations, with pooled adjusted ORs of 1.75 (95% CI: [1.54, 1.99]) and 1.70 (95% CI: [1.55, 1.86]), respectively. Substantial variability among countries

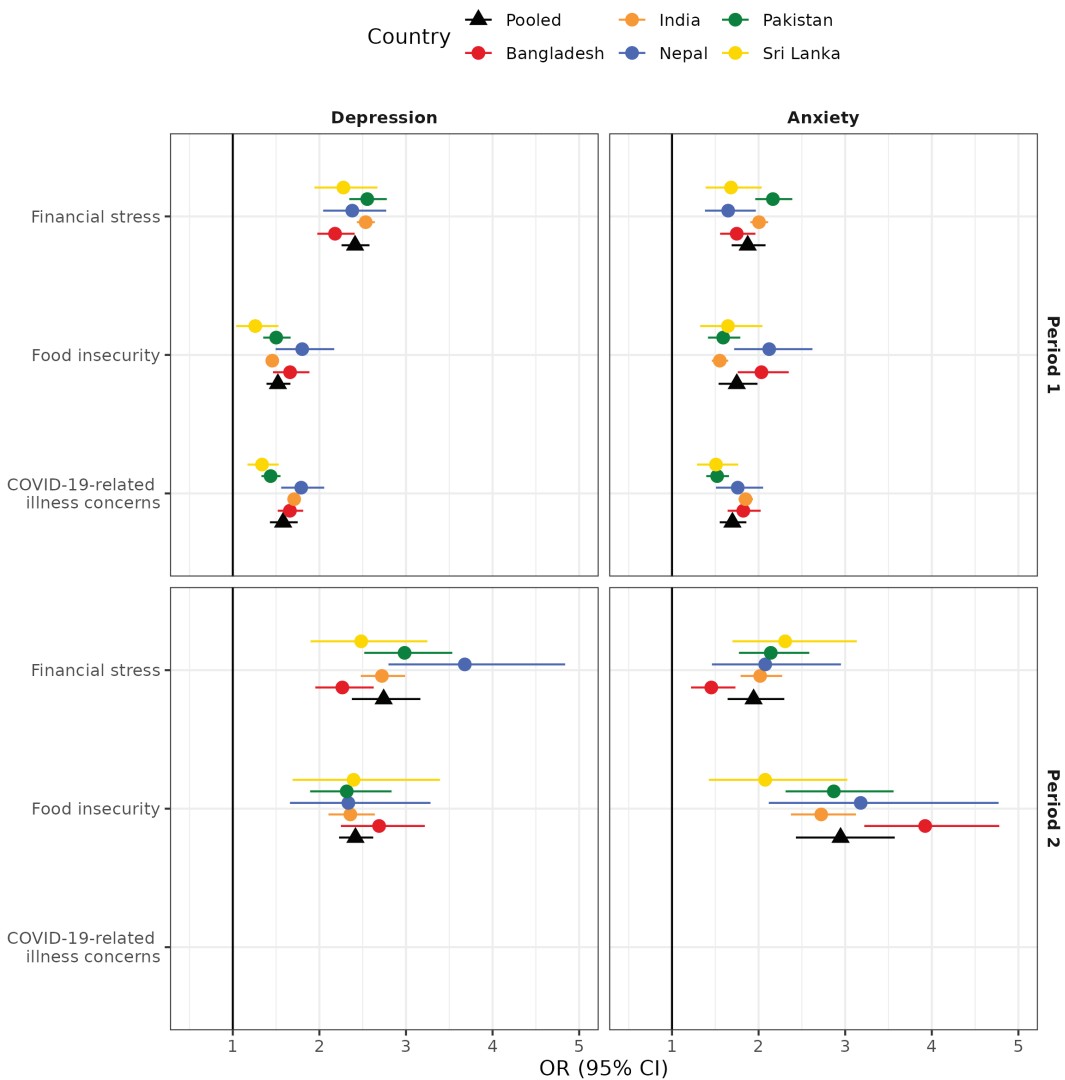

**Fig 1. Unpooled (circles) and pooled (triangles) effects of pandemic-related worries on mental health across five South Asian countries in Period 1 (N = 827,472) and Period 2 (N = 235,314), post-weighting.** Abbreviations: Period 1, June 27, 2020, to May 19, 2021; Period 2, May 20, 2021, to June 25, 2022; OR, odds ratio; CI, confidence interval. COVID-19-related health concerns were excluded from the surveys during Period 2. Separate weighted logistic regression models were fitted for complete cases stratified by each period and country, including pandemic-related worries (financial stress, food insecurity, and COVID-19-related illness concerns), demographics (gender, age, education, rural-urban residential status, and occupation), and calendar time (categorized by month and year) as covariates. The pooled estimates were obtained by a random-effects meta-analysis. The results present odds ratios with their corresponding 95% Wald confidence intervals. Robust sandwich estimators were applied for variance estimation. The vertical solid black lines represent an odds ratio of 1.

was evident, with heterogeneity ($I^2$) values of 77.22% for financial stress, 78.53% for food insecurity, and 77.01% for COVID-19-related illness concerns.

Country-specific analyses in Period 1 illustrated distinct patterns. In Pakistan, anxiety was most strongly associated with financial stress (adjusted OR = 2.17; 95% CI: [1.96, 2.39]), followed by food insecurity (adjusted OR = 1.59; 95% CI: [1.42, 1.79]) and COVID-19-related illness concerns (adjusted OR = 1.52; 95% CI: [1.40, 1.66]). A similar association pattern was evident in India but was less pronounced or absent in the other studied countries.

In Period 2, pooled estimates indicated a shift, with food insecurity becoming the predominant factor associated with anxiety, reflected by a substantially higher pooled adjusted OR of 2.95 (95% CI: [2.43, 3.57]), compared to financial stress (adjusted OR = 1.94; 95% CI: [1.64, 2.30]). Variability across studies remained substantial, with $I^2$ values of 70.26% for financial stress and 69.83% for food insecurity.

Country-specific analyses in Period 2 emphasized the elevated importance of food insecurity as a driver of anxiety. In Bangladesh, food insecurity exhibited a notably high adjusted OR of 3.92 (95% CI: [3.22, 4.78]), markedly greater than the association with financial stress (adjusted OR = 1.45; 95% CI: [1.22, 1.73]). Similarly, in India, the association between food insecurity and anxiety intensified notably from an adjusted OR of 1.55 (95% CI: [1.46, 1.65]) in Period 1 to 2.72 (95% CI: [2.37, 3.12]) in Period 2.

## Effect modifiers of associations between pandemic-related worries and mental health

**Calendar time.**   We assessed how the associations between three pandemic-related worries and two mental health measures evolved over calendar time (categorized by month and year) within each period, based on the complete cases (N = 1,062,786). Fig 2 depicts the adjusted ORs and 95% CIs for these associations in Bangladesh across time in both periods.

During Period 2, the adjusted ORs for anxiety linked to food insecurity consistently exceeded those linked to financial stress. Specifically, adjusted ORs regarding food insecurity ranged from 2.13 (95% CI: [1.42, 3.19]) to 6.94 (95% CI: [3.60, 13.40]), compared to those regarding financial stress, which ranged from 0.91 (95% CI: [0.56, 1.47]) to 2.40 (95% CI: [1.41, 4.08]). Notably, the associations between financial stress and anxiety lacked sufficient evidence to reach statistical significance from May to September 2021, January to March 2022, and June 2022. In contrast, food insecurity showed no overlap in CIs with financial stress during specific months; for example, in October 2021, the adjusted OR regarding food insecurity was 5.13 (95% CI: [3.16, 8.31]), compared to 1.67 (95% CI: [1.08, 2.57]) regarding financial stress. Similar patterns were observed in January 2022, March 2022, May 2022, and June 2022.

Results for India and Pakistan are provided in S2 Figure and S3 Figure. Analyses for Nepal and Sri Lanka were not shown because the limited monthly sample sizes (fewer than 2,000 respondents) resulted in considerable uncertainty.

**Vaccination status.**   We examined whether vaccination status modified the association between pandemic-related worries and mental health during Period 2 across five South Asian countries, using complete cases with no missing vaccination data (N = 234,149). Fig 3 presents the adjusted ORs and 95% CIs for these associations across these countries between different vaccination statuses, stratified by outcomes and worry variables.

For both depression and anxiety, the adjusted ORs for financial stress and food insecurity were generally higher among individuals who had received at least one dose of the COVID-19 vaccine compared to those who were unvaccinated, particularly in Bangladesh, India, Nepal, and Sri Lanka, after adjusting for demographics and calendar time. For instance, in Bangladesh, the adjusted OR for financial stress on depression among vaccinated individuals was 2.88 (95% CI: [2.61, 3.19]) compared to 1.82 (95% CI: [1.53, 2.17]) among unvaccinated individuals. The p-value for the interaction between vaccination status and financial stress (denoted as $p_{int}$) was less than 0.01, indicating that the association between financial stress and depression significantly differed by vaccination status. In Sri Lanka, we found insufficient evidence of a significant association between financial stress and depression among

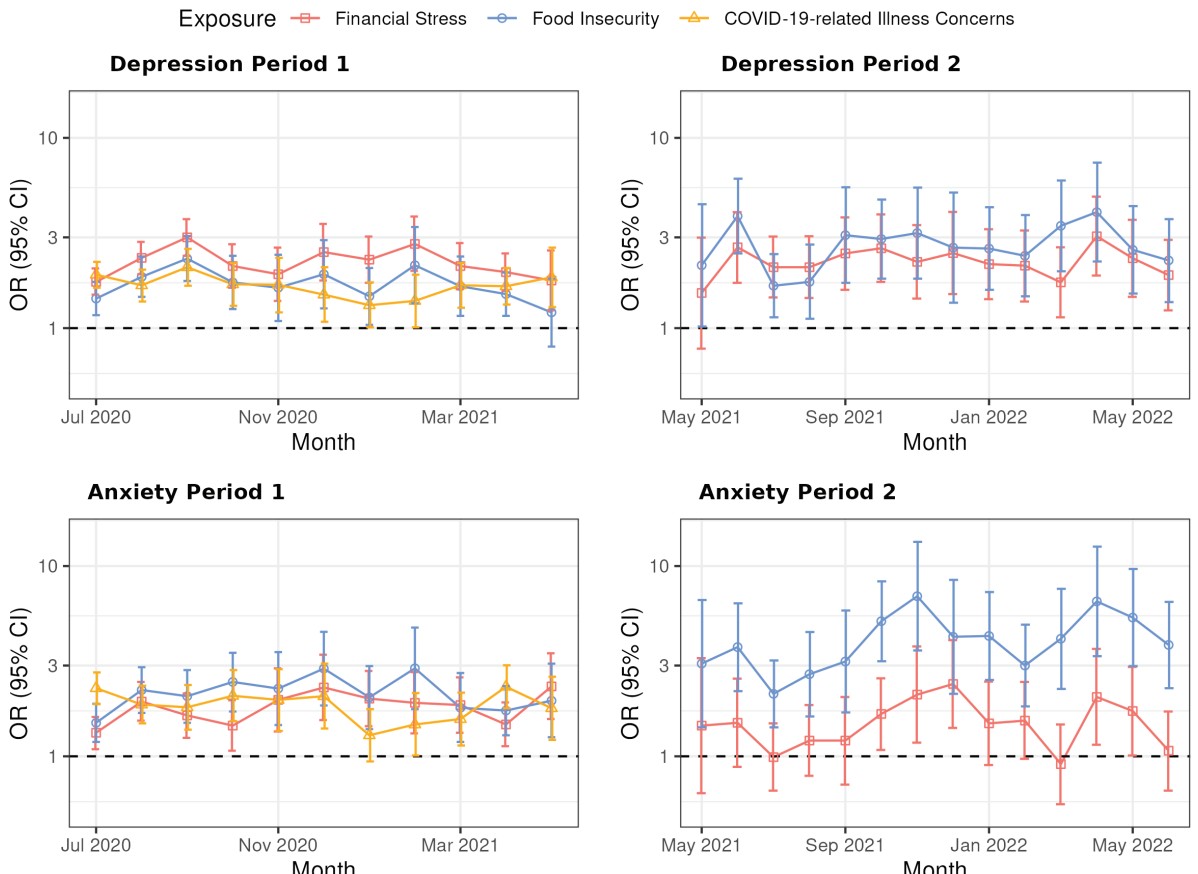

**Fig 2. Effects of pandemic-related worries on mental health over time in Bangladesh during Period 1 (N = 69,047) and Period 2 (N = 34,004), post-weighting.** Abbreviations: Period 1, June 27, 2020, to May 19, 2021; Period 2, May 20, 2021, to June 25, 2022; OR, odds ratio; CI, confidence interval. COVID-19-related health concerns were excluded from the surveys during Period 2. Separate weighted logistic regression models were fitted for complete cases from each period, including pandemic-related worries (financial stress, food insecurity, and COVID-19-related illness concerns), demographics (gender, age, education, rural-urban residential status, and occupation), and calendar time (categorized by month and year) as covariates. Interaction terms between time and one worry variable were included in each model. The results present odds ratios with their corresponding 95% Wald confidence intervals. Robust sandwich estimators were applied for variance estimation. The y-axis is displayed on a logarithmic scale to improve visualization. The horizontal dashed black lines represent an odds ratio of 1.

unvaccinated individuals (adjusted OR 1.52 (95% CI:[0.91, 2.54])), while the association was significant among vaccinated individuals (adjusted OR 2.79 (95% CI:[2.10, 3.71])). In contrast, we found insufficient evidence that vaccination status modified the associations between pandemic-related worries and mental health in Pakistan, as all interaction term p-values exceeded 0.05.

**Self-reported gender.** We investigated whether the associations between pandemic-related worries and both depression and anxiety differed by gender, based on complete cases (N = 1,062,786). S4 Figure shows the adjusted ORs and 95% CIs for these associations, stratified by gender, mental health outcomes, periods, and worry variables across countries.

In Period 1 in India, financial stress and food insecurity were significantly more strongly associated with both depression and anxiety among males than females. For example, the adjusted OR for anxiety linked to food insecurity was 1.58 (95% CI: [1.51, 1.66]) among males compared to 1.18 (95% CI: [1.05, 1.32]) among females, with a significant interaction term

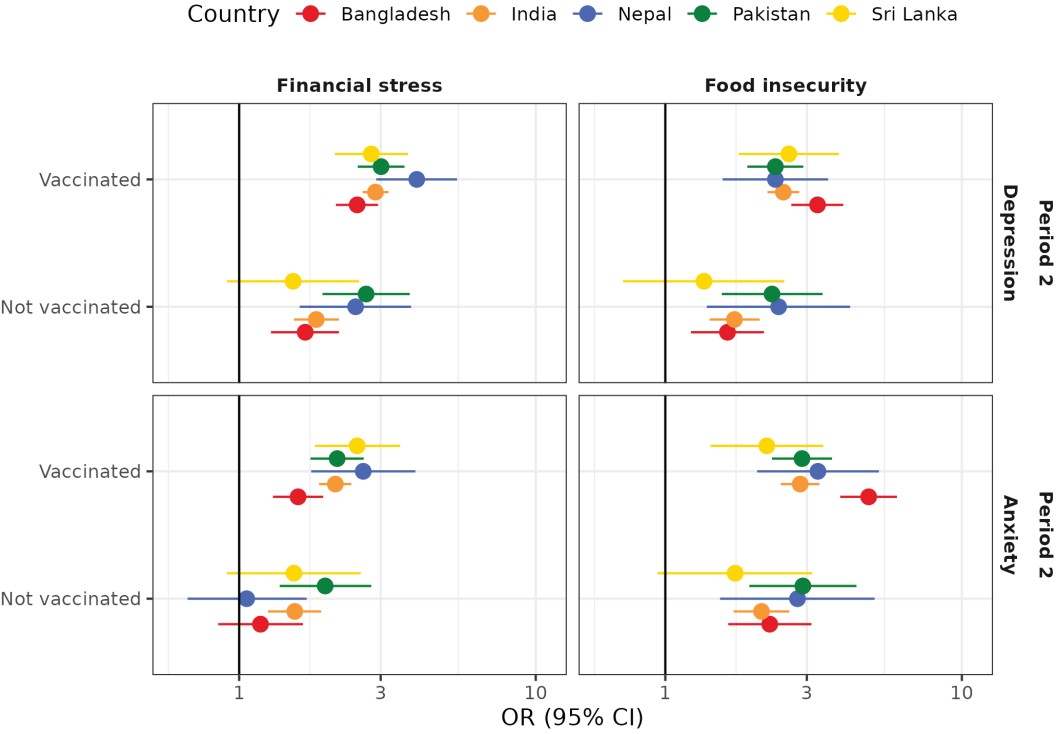

**Fig 3. Effects of pandemic-related worries on mental health between vaccination statuses across five South Asian countries during Period 2 (N = 234,149), post-weighting.** Abbreviations: Period 2, May 20, 2021, to June 25, 2022; OR, odds ratio; CI, confidence interval. COVID-19-related health concerns were excluded from the surveys during Period 2. Separate weighted logistic regression models were fitted for complete cases from each period, including pandemic-related worries (financial stress, food insecurity, and COVID-19-related illness concerns), demographics (gender, age, education, rural-urban residential status, occupation, and vaccination status), and calendar time (categorized by month and year) as covariates. Interaction terms between vaccination status and one worry variable were included in each model. The results present odds ratios with their corresponding 95% Wald confidence intervals. Robust sandwich estimators were applied for variance estimation. The x-axis is presented on a logarithmic scale to improve visualization. The vertical black lines at an odds ratio of 1 represent the null.

($p_{int}$ < 0.01). Similar gender-based differences were observed in Pakistan, where food insecurity was more strongly associated with both depression and anxiety among males in Period 1. In Bangladesh, stronger associations between food insecurity and depression were observed among males in Period 1, as well as between financial stress and depression in both periods.

Conversely, in Sri Lanka, females had a significantly stronger association between financial stress and anxiety in Period 2 compared to males ($p_{int}$ = 0.02), with an adjusted OR of 3.24 (95% CI: [2.05, 5.11]) for females and 1.71 (95% CI: [1.24, 2.34]) for males.

**Education.** We explored whether education was an effect modifier on the associations between pandemic-related worries and mental health in a complete case analysis (N = 1,062,786). S5 Figure displays the adjusted ORs and 95% CIs for these associations across countries, stratified by education, mental health outcomes, periods, and worry variables.

In India, across both periods, individuals with a high school education or above showed significantly stronger associations between the three pandemic-related worries (when available) and mental health outcomes compared to those with lower educational attainment.

For example, in Period 1, the adjusted OR for depression related to COVID-19-related illness concerns was 1.78 (95% CI: [1.71, 1.85]) among individuals with a high school education or more, versus 1.47 (95% CI: [1.36, 1.58]) among those with less ($p_{int} < 0.01$). Similar trends occurred in Nepal for the associations between financial stress and both mental health outcomes during Period 1 and in Sri Lanka for the association between food insecurity and anxiety during Period 1.

By contrast, in Sri Lanka during Period 2, individuals without a high school degree had a significantly stronger association between financial stress and anxiety compared to those with at least a high school education ($p_{int} = 0.02$). The adjusted OR for anxiety linked to financial stress was 4.82 (95% CI: [2.19, 10.62]) among participants without a high school education, compared to 1.77 (95% CI: [1.16, 2.68]) among those with a high school degree or more.

**Rural-urban residential status.** We evaluated whether rural-urban residential status modified the associations between pandemic-related worries and mental health using the complete cases (N = 1,062,786). S6 Figure illustrates the adjusted ORs and 95% CIs for these associations across countries, stratified by residential status, mental health outcomes, periods, and worry variables.

Consistent with our findings regarding education, in India, urban residents demonstrated significantly stronger associations between the three pandemic-related worries (when available) and mental health outcomes compared to rural residents. For instance, during Period 2, the adjusted OR for anxiety associated with food insecurity was 2.97 (95% CI: [2.52, 3.50]) among urban individuals, compared to 2.39 (95% CI: [2.01, 2.83]) among rural residents ($p_{int} < 0.01$). Similar patterns were noted in Pakistan for the associations between food insecurity and anxiety in Period 1 and between COVID-19-related illness concerns and anxiety in Period 1, as well as in Sri Lanka for the associations between food insecurity and anxiety in Period 2 and financial stress and anxiety in Period 2.

## Discussion

### How do our findings relate to the literature during non-pandemic times?

Our primary finding indicates that all three worries—financial stress, food insecurity, and COVID-19-related illness concerns—were significantly associated with increased levels of depression and anxiety in all five countries, after adjusting for demographics and calendar time. This pattern aligns with the pre-pandemic research linking economic and health-related stressors to poor mental health outcomes [2–4].

Importantly, our study is among the first to examine the role of socioeconomic factors as potential effect modifiers of these associations in LMIC settings. We found notable heterogeneity in how worries influenced mental health across different socioeconomic subgroups and countries. For example, in India, these worries were more strongly associated with frequent self-reported depression and anxiety among males, individuals with at least a high school education, and urban residents compared to their respective counterparts. Conversely, in Sri Lanka, we observed the opposite trend, with weaker associations in similar subgroups. These cross-country differences in effect modification are broadly consistent with the limited literature outside the pandemic context. Tran et al. [5] found that financial stress was more strongly linked to anxiety among female college students in the United States, while other studies reported no significant effect modification by gender [42–44] and similarly, no modifying role of education [42,44]. The heterogeneity in effect modification across countries underscores the limitations of one-size-fits-all approaches to mental health programming.

## How do our findings compare with global research during the pandemic?

Our findings demonstrating significant associations between pandemic-related worries and poorer mental health outcomes align with previous research conducted outside South Asia during the COVID-19 pandemic [15,21,25].

In four of the five South Asian countries examined, vaccination status was associated with improved mental health outcomes—mirroring U.S. findings of reduced distress [66,67] (see S2 Table)). Intriguingly, vaccination status also emerged as an effect modifier of the relationship between pandemic-related worries and adverse mental health outcomes. Specifically, while vaccinated individuals generally reported lower levels of depression and anxiety, the strength of association between their worry and mental health was greater than among unvaccinated individuals. One plausible explanation for this paradox is reverse causation: those who opted for vaccination may have initially had higher levels of depression or anxiety, motivating their decision to seek vaccination. Another possibility is the expectation hypothesis—vaccination may have been widely perceived as a definitive pathway back to normalcy. Consequently, vaccinated individuals might have experienced a greater sense of disappointment when pandemic disruptions continued despite vaccination. To our knowledge, this dual role of vaccination—simultaneously protective as a direct factor against depression and anxiety, yet amplifying the sensitivity of mental health outcomes to pandemic-related worries—is a novel finding.

## How do our findings compare with pandemic-era studies in South Asia?

The availability of CTIS data across multiple South Asian countries over an extended period offers a rare opportunity to examine mental health outcomes within a unified study framework. Unlike meta-analyses that synthesize findings from separate country-level studies [30]—each with potentially different designs and measurement tools—our analysis benefits from a consistent methodology and survey instrument, thereby reducing heterogeneity due to study design or measurement variation.

Our findings on the significant associations between pandemic-related worries and poorer mental health in South Asia are broadly consistent with the limited existing literature in the region, which has predominantly focused on specific subgroups [37–41].

In our study, we observed that the adjusted ORs for depression and anxiety associated with pandemic-related worries varied across the five countries, with differences in the rankings of adjusted ORs among the different worry variables. These variations reflect underlying differences in pre-pandemic socio-economic structures and the coping strategies adopted by each country during the pandemic. For instance, the five countries implemented distinct lockdown policies [32], which have been shown to impact mental health in the general population [68]. Such differences in public health responses and social systems may have influenced the mental health burden in each country, highlighting the complicated interplay between policy, socio-economic resilience, and mental well-being [69].

We also noticed changes in these associations over time. Using Bangladesh as a case study, we identified a more pronounced effect of worries about food insecurity on anxiety during Period 2 (May 20, 2021, to June 25, 2022), compared to financial stress. Period 2 was marked by a sharp rise in the prices of essential commodities in Bangladesh, including food and fuel—particularly wheat and palm oil—with prices escalating further after February 2022 [70]. This surge can largely be attributed to disruptions in the global supply chain caused by the Russia-Ukraine war [71]. These findings highlight the need for careful interpretation, as factors beyond the direct impact of COVID-19, such as geopolitical conflicts, can shape the relationship between pandemic-related stressors and mental health over time.

## Limitations

This study has several limitations. First, the use of a non-probabilistic sample distributed via a social media platform introduces inherent selection bias due to participants' self-selection [72]. Despite the weighting adjustments provided by Facebook, this bias could not be fully corrected, particularly for variables not included in the weighting process. For example, the sample consistently overrepresented individuals with a high school education or higher, as well as those living in urban areas across all five countries, diverging from national census benchmarks. This limitation, observed in prior studies as well [50,73], may restrict the generalizability of our findings to the broader adult population. Second, both depression and anxiety were assessed using self-reported symptom frequencies, which do not equate to clinical diagnoses. Demographic variables were also self-reported, introducing the possibility of reporting bias, measurement error, and misclassification. Third, the discontinuation of questions regarding COVID-19-related illness concerns on May 20, 2021, prevented us from assessing the impact of this worry variable on mental health during Period 2.

Future research would benefit from using probabilistic sampling to improve representativeness and provide more robust mental health assessments. Incorporating longitudinal and primary data, alongside social media-based survey data, would also allow for a more comprehensive understanding of how public health response measures impact mental health during health emergencies.

## Conclusion

To our knowledge, this is the first comparative study in South Asia to assess the impact of pandemic-related worries on mental health in the general public during COVID-19 from 2020 to 2022 using the same survey and deployment platform (CTIS/Meta). It extends our previous work in India [50] to a broader regional analysis and highlights the similarities and differences across South Asia. This research is particularly important given the absence of national-level monitoring systems for longitudinal tracking of mental health symptoms across countries in this region during the pandemic [74]. While most of the literature pool results from disparate studies via meta-analysis, this study uses a common global survey, reducing measurement and selection heterogeneity across countries.

Our findings indicate significant associations between pandemic-related worries and self-reported depression and anxiety. These results suggest that enhancing economic stability and ensuring food security are essential strategies for mitigating the mental health impacts of prolonged crises such as the COVID-19 pandemic. Addressing these socioeconomic factors alongside expanding access to professional mental health support may help alleviate the burden on affected populations [52].

Moreover, our analysis revealed considerable variation in these associations across the five South Asian countries, with differences observed by outcome, calendar time, and individual characteristics. This heterogeneity underscores the need for context-specific, time- and location-adjusted policies that effectively address the unique challenges faced by each population, particularly those most severely impacted by the pandemic. This paper contributes to the literature by addressing the largely unexplored role of effect modifiers, including that of time and vaccination status, on the association between pandemic-related worries and mental health.

Finally, our study highlights the importance of establishing real-time monitoring systems for public mental health. Such systems could inform daily interventions and enhance preparedness for future crises.

## Supporting information

**S1 Text. Covariates.** Data processing for demographic variables.
(PDF)

**S2 Text. Statistical models.** Detailed mathematical descriptions of the statistical models used in the analysis.
(PDF)

**S1 Table. Question phrasing and initial answers in the survey for Period 1 and Period 2.**
Abbreviations: Period 1, June 27, 2020, to May 19, 2021; Period 2, May 20, 2021, to June 25, 2022. COVID-19-related health concerns were excluded from the surveys during Period 2. Note: a) The survey question on vaccination status was added on January 5, 2021.
(PDF)

**S2 Table. Effects of vaccination status on mental health and pandemic-related worries across five South Asian countries during Period 2 (N = 234,149), post-weighting.** Abbreviations: Period 2, May 20, 2021, to June 25, 2022. The results were obtained from separate unadjusted models that include vaccination status as the only covariate per model. All model types integrate survey weights within logistic regression. Only complete cases with no missing data on vaccination status, demographics, pandemic-related worries, and outcomes were included in the analysis. Odds ratios are displayed as estimates with corresponding 95% Wald confidence intervals in the form of estimates [95% confidence interval], using a robust sandwich estimator for variance calculation. Significant odds ratios from Wald tests (significance level: 0.05) are highlighted in bold.
(PDF)

**S1 Fig. Overview of data processing.** This secondary analysis draws on data from the COVID-19 Trends and Impact Survey, a global online survey administered to adult Facebook active users. The study period spans from June 27, 2020, to June 25, 2022, with a major survey revision on May 20, 2021, marking two distinct analytical phases. Descriptive analyses include the full sample from five South Asian countries (Bangladesh, India, Nepal, Pakistan, and Sri Lanka), while statistical analyses are limited to respondents with complete data.
(PDF)

**S2 Fig. Effects of pandemic-related worries on mental health over time in India during Period 1 (N = 595,229) and Period 2 (N = 152,767), post-weighting.** Abbreviations: Period 1, June 27, 2020, to May 19, 2021; Period 2, May 20, 2021, to June 25, 2022; OR, odds ratio; CI, confidence interval. COVID-19-related health concerns were excluded from the surveys during Period 2. Separate weighted logistic regression models were fitted for complete cases from each period, including pandemic-related worries (financial stress, food insecurity, and COVID-19-related illness concerns), demographics (gender, age, education, rural-urban residential status, and occupation), and calendar time (categorized by month and year) as covariates. Interaction terms between time and one worry variable were included in each model. The results present odds ratios with their corresponding 95% Wald confidence intervals. Robust sandwich estimators were applied for variance estimation. The y-axis is displayed on a logarithmic scale to improve visualization. The horizontal dashed black lines represent an odds ratio of 1.
(PDF)

**S3 Fig. Effects of pandemic-related worries on mental health over time in Pakistan during Period 1 (N = 90,090) and Period 2 (N = 28,971), post-weighting.** Abbreviations: Period 1, June 27, 2020, to May 19, 2021; Period 2, May 20, 2021, to June 25, 2022; OR, odds ratio;

CI, confidence interval. COVID-19-related health concerns were excluded from the surveys during Period 2. Separate weighted logistic regression models were fitted for complete cases from each period, including pandemic-related worries (financial stress, food insecurity, and COVID-19-related illness concerns), demographics (gender, age, education, rural-urban residential status, and occupation), and calendar time (categorized by month and year) as covariates. Interaction terms between time and one worry variable were included in each model. The results present odds ratios with their corresponding 95% Wald confidence intervals. Robust sandwich estimators were applied for variance estimation. The y-axis is displayed on a logarithmic scale to improve visualization. The horizontal dashed black lines represent an odds ratio of 1.
(PDF)

**S4 Fig. Effects of pandemic-related worries on mental health between gender across five South Asian countries in Period 1 (N = 827,472) and Period 2 (N = 235,314), post-weighting.** Abbreviations: Period 1, June 27, 2020, to May 19, 2021; Period 2, May 20, 2021, to June 25, 2022; OR, odds ratio; CI, confidence interval. COVID-19-related health concerns were excluded from the surveys during Period 2. Separate weighted logistic regression models were fitted for complete cases from each period, including pandemic-related worries (financial stress, food insecurity, and COVID-19-related illness concerns), demographics (gender, age, education, rural-urban residential status, and occupation), and calendar time (categorized by month and year) as covariates. Interaction terms between gender and one worry variable were included in each model. The results present odds ratios with their corresponding 95% Wald confidence intervals. Robust sandwich estimators were applied for variance estimation. The x-axis is presented on a logarithmic scale to improve visualization. The vertical black lines at an odds ratio of 1 represent the null.
(PDF)

**S5 Fig. Effects of pandemic-related worries on mental health between education across five South Asian countries in Period 1 (N = 827,472) and Period 2 (N = 235,314), post-weighting.** Abbreviations: Period 1, June 27, 2020, to May 19, 2021; Period 2, May 20, 2021, to June 25, 2022; OR, odds ratio; CI, confidence interval; HS, high school. COVID-19-related health concerns were excluded from the surveys during Period 2. Separate weighted logistic regression models were fitted for complete cases from each period, including pandemic-related worries (financial stress, food insecurity, and COVID-19-related illness concerns), demographics (gender, age, education, rural-urban residential status, and occupation), and calendar time (categorized by month and year) as covariates. Interaction terms between education and one worry variable were included in each model. The results present odds ratios with their corresponding 95% Wald confidence intervals. Robust sandwich estimators were applied for variance estimation. The x-axis is presented on a logarithmic scale to improve visualization. The vertical black lines at an odds ratio of 1 represent the null.
(PDF)

**S6 Fig. Effects of pandemic-related worries on mental health between rural-urban residential statuses across five South Asian countries in Period 1 (N = 827,472) and Period 2 (N = 235,314), post-weighting.** Abbreviations: Period 1, June 27, 2020, to May 19, 2021; Period 2, May 20, 2021, to June 25, 2022; OR, odds ratio; CI, confidence interval. COVID-19-related health concerns were excluded from the surveys during Period 2. Separate weighted logistic regression models were fitted for complete cases from each period, including pandemic-related worries (financial stress, food insecurity, and COVID-19-related illness concerns), demographics (gender, age, education, rural-urban residential status, and

occupation), and calendar time (categorized by month and year) as covariates. Interaction terms between rural-urban residential statuses and one worry variable were included in each model. The results present odds ratios with their corresponding 95% Wald confidence intervals. Robust sandwich estimators were applied for variance estimation. The x-axis is presented on a logarithmic scale to improve visualization. The vertical black lines at an odds ratio of 1 represent the null.
(PDF)

## Acknowledgments

We sincerely thank the Social Data Science Center at the University of Maryland and Meta/Facebook for their pivotal role in conducting the global CTIS and for providing access to the anonymized individual-level data.

## Author contributions

**Conceptualization:** Bhramar Mukherjee.

**Data curation:** Youqi Yang, Bhramar Mukherjee.

**Formal analysis:** Youqi Yang, Bhramar Mukherjee.

**Funding acquisition:** Bhramar Mukherjee.

**Investigation:** Youqi Yang, Bhramar Mukherjee.

**Methodology:** Youqi Yang, Bhramar Mukherjee.

**Software:** Youqi Yang.

**Supervision:** Bhramar Mukherjee.

**Visualization:** Youqi Yang.

**Writing – original draft:** Youqi Yang, Bhramar Mukherjee.

**Writing – review & editing:** Youqi Yang, Lauren Zimmermann, Santanu Pramanik, Brian Wahl, Bhramar Mukherjee.

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
