## [Decision Letter · Decision Letter 0]

29 Apr 2025

PGPH-D-25-00676

The role of financial stress, food insecurity, and COVID-19-related illness concerns shaping mental health in South Asia during the pandemic years (2020–2022)

Dear Dr. Youqi Yang,

Thank you for submitting your manuscript to PLOS Global Public Health. After careful consideration, we feel that it has merit but does not fully meet PLOS Global Public Health’s publication criteria as it currently stands. Therefore, we invite you to submit a revised version of the manuscript that addresses the points raised during the review process.

We look forward to receiving your revised manuscript.

Kind regards,

Muhammad Asaduzzaman, MD MPH MPhil

Academic Editor

Journal Requirements:

1. We ask that a manuscript source file is provided at Revision. Please upload your manuscript file as a .doc, .docx, .rtf or .tex.

Additional Editor Comments (if provided):

Reviewers' comments:

Reviewer's Responses to Questions

**Comments to the Author**

1. Does this manuscript meet PLOS Global Public Health’s publication criteria? Is the manuscript technically sound, and do the data support the conclusions? The manuscript must describe methodologically and ethically rigorous research with conclusions that are appropriately drawn based on the data presented.

Reviewer #1: Yes

Reviewer #2: Yes

2. Has the statistical analysis been performed appropriately and rigorously?

Reviewer #1: Yes

Reviewer #2: Yes

3. Have the authors made all data underlying the findings in their manuscript fully available (please refer to the Data Availability Statement at the start of the manuscript PDF file)?

Reviewer #1: Yes

Reviewer #2: No

4. Is the manuscript presented in an intelligible fashion and written in standard English?

Reviewer #1: Yes

Reviewer #2: Yes

5. Review Comments to the Author

Reviewer #1: The topic is highly relevant, and the authors present a valuable analysis. However, the manuscript titled "The role of financial stress, food insecurity, and COVID-19-related illness concerns shaping mental health in South Asia during the pandemic years (2020–2022)" (PGPH-D-25-00676_reviewer) requires minor revisions before it can be considered for publication. (Kindly see attached file)

Reviewer #2: Dear Author,

Thank you for your work and for submitting this manuscript. I have a few comments that I hope will help strengthen your paper:

Title Clarity: Could the title more clearly reflect the type of study conducted? I had to read into the manuscript to determine the study design. Including this information in the title could improve clarity and accessibility for readers.

Lines 36–40 Pandemic-Related Worries: The connection between financial stress, food insecurity, and the pandemic is not well established in the current manuscript. These issues certainly exist independently of the pandemic. If the intent is to highlight that these concerns worsened or became more prominent due to the pandemic, please provide supporting evidence or references to substantiate this claim.

Line 46 Online Survey and Response Bias: You mention the use of an online survey, yet this does not appear in your limitations section. Given that the study population includes a substantial proportion of rural and potentially lower-educated individuals, this raises concerns about response bias and representativeness. How was this addressed or mitigated? If you have data or references indicating that mobile phone and internet usage in the surveyed areas is sufficient to support representative online data collection, please include this to justify your methodology.

Figure 3 Contradictory Findings: Figure 3 suggests higher odds ratios for outcome variables among vaccinated and urban individuals, which contrasts with the narrative in lines 434–438. This discrepancy is not clearly addressed in the text. Could you elaborate on this or provide a potential explanation?

Thank you again for your contributions, and I look forward to seeing your revised manuscript.

6. PLOS authors have the option to publish the peer review history of their article (what does this mean?). If published, this will include your full peer review and any attached files.

**Do you want your identity to be public for this peer review?** For information about this choice, including consent withdrawal, please see our Privacy Policy.

Reviewer #1: No

Reviewer #2: **Yes: **Isaac Che Ngang

---

## [Decision Letter · Decision Letter 1]

9 Jul 2025

PGPH-D-25-00676R1

The role of financial stress, food insecurity, and COVID-19-related illness concerns shaping mental health in five South Asian countries during the pandemic (2020–2022): A secondary analysis of the online COVID-19 Trends and Impact Survey (CTIS) data

Dear Dr. Youqi Yang,

Thank you for submitting your manuscript to PLOS Global Public Health. After careful consideration, we feel that it has merit but does not fully meet PLOS Global Public Health’s publication criteria as it currently stands. Therefore, we invite you to submit a revised version of the manuscript that addresses the points raised during the review process.

We look forward to receiving your revised manuscript.

Kind regards,

Muhammad Asaduzzaman, MD MPH MPhil

Academic Editor

Journal Requirements:

Additional Editor Comments (if provided):

Reviewers' comments:

Reviewer's Responses to Questions

**Comments to the Author**

1. If the authors have adequately addressed your comments raised in a previous round of review and you feel that this manuscript is now acceptable for publication, you may indicate that here to bypass the “Comments to the Author” section, enter your conflict of interest statement in the “Confidential to Editor” section, and submit your "Accept" recommendation.

Reviewer #1: All comments have been addressed

Reviewer #2: All comments have been addressed

2. Does this manuscript meet PLOS Global Public Health’s publication criteria? Is the manuscript technically sound, and do the data support the conclusions? The manuscript must describe methodologically and ethically rigorous research with conclusions that are appropriately drawn based on the data presented.

Reviewer #1: Partly

Reviewer #2: Yes

3. Has the statistical analysis been performed appropriately and rigorously?

Reviewer #1: Yes

Reviewer #2: Yes

4. Have the authors made all data underlying the findings in their manuscript fully available (please refer to the Data Availability Statement at the start of the manuscript PDF file)?

Reviewer #1: Yes

Reviewer #2: Yes

5. Is the manuscript presented in an intelligible fashion and written in standard English?

Reviewer #1: Yes

Reviewer #2: Yes

6. Review Comments to the Author

Reviewer #1: Title: The Role of Financial Stress, Food Insecurity, and COVID-19-Related Illness Concerns in Shaping Mental Health in Five South Asian Countries During the Pandemic (2020–2022): A Secondary Analysis of the Online COVID-19 Trends and Impact Survey (CTIS) Data

Manuscript Number: PGPH-D-25-00676R1

Reviewer Comments:

The topic is interesting and well appreciated. Kindly revise the manuscript before it is considered for publication.

Abstract:

This section is well written.

Introduction:

This section still requires improvement. Please ensure a logical flow based on the study objectives. Develop the section by incorporating relevant previous studies as part of the literature review. Clearly state the problem and rationale, explaining why this study is timely and necessary.

The following lines should be relocated to the Materials and Methods section:

"Our study used data from the COVID-19 Trends and Impact Survey (CTIS), which was conducted on the Facebook social media platform in collaboration with Meta/Facebook, Carnegie Mellon University (for the United States) [45], and the University of Maryland (globally) [46]. This survey, with its broad coverage of COVID-19-related topics, has proven to be a valuable resource for understanding mental health [47–49] and its associations with pandemic-related worries [50, 51]."

Materials and Methods:

Please elaborate on how the number of respondents (N = 3,644,631) was determined and how they were distributed across the five countries. What were the inclusion criteria for respondents? Also, describe in detail the process of using the Facebook platform for data collection.

Results:

This section is well written.

Discussion:

This section needs further improvement. Please strengthen it by justifying your findings with relevant and recent studies.

Conclusion:

Kindly elaborate on the study’s contribution to the existing body of literature.

Citations and References:

This section is well organized.

General Comments:

Please remove repetition and redundancy, and improve the overall language quality. Ensure a logical flow throughout the manuscript that aligns with the objectives.

Reviewer #2: Thank you for addressing the feedback provided. The revisions have significantly strengthened the manuscript, and we appreciate your efforts in improving its clarity and rigor.

7. PLOS authors have the option to publish the peer review history of their article (what does this mean?). If published, this will include your full peer review and any attached files.

**Do you want your identity to be public for this peer review?** For information about this choice, including consent withdrawal, please see our Privacy Policy.

Reviewer #1: No

Reviewer #2: No

---

## [Editor Report · Decision Letter 2]

22 Jul 2025

The role of financial stress, food insecurity, and COVID-19-related illness concerns shaping mental health in five South Asian countries during the pandemic (2020–2022): A secondary analysis of the online COVID-19 Trends and Impact Survey (CTIS) data

PGPH-D-25-00676R2

Dear Youqi Yang

We are pleased to inform you that your manuscript 'The role of financial stress, food insecurity, and COVID-19-related illness concerns shaping mental health in five South Asian countries during the pandemic (2020–2022): A secondary analysis of the online COVID-19 Trends and Impact Survey (CTIS) data' has been provisionally accepted for publication in PLOS Global Public Health.

Best regards,

Muhammad Asaduzzaman, MD MPH MPhil

Academic Editor